# Vulnerability of the North Water ecosystem to climate change

Sofia Ribeiro [1✉], Audrey Limoges [1,2], Guillaume Massé[3,4], Kasper L. Johansen [5], William Colgan [1], Kaarina Weckström [1,6], Rebecca Jackson [1], Eleanor Georgiadis[3,7], Naja Mikkelsen[1], Antoon Kuijpers [1], Jesper Olsen [8], Steffen M. Olsen [9], Martin Nissen[10], Thorbjørn J. Andersen [11], Astrid Strunk [12], Sebastian Wetterich [13], Jari Syväranta[14], Andrew C. G. Henderson [15], Helen Mackay[15,16], Sami Taipale [17], Erik Jeppesen[18,19,20], Nicolaj K. Larsen[12,21], Xavier Crosta [7], Jacques Giraudeau[7], Simone Wengrat[22], Mark Nuttall[23,24], Bjarne Grønnow[25], Anders Mosbech [5] & Thomas A. Davidson [18✉]

High Arctic ecosystems and Indigenous livelihoods are tightly linked and exposed to climate change, yet assessing their sensitivity requires a long-term perspective. Here, we assess the vulnerability of the North Water polynya, a unique seaice ecosystem that sustains the world's northernmost Inuit communities and several keystone Arctic species. We reconstruct mid-to-late Holocene changes in sea ice, marine primary production, and little auk colony dynamics through multi-proxy analysis of marine and lake sediment cores. Our results suggest a productive ecosystem by 4400–4200 cal yrs b2k coincident with the arrival of the first humans in Greenland. Climate forcing during the late Holocene, leading to periods of polynya instability and marine productivity decline, is strikingly coeval with the human abandonment of Greenland from c. 2200–1200 cal yrs b2k. Our long-term perspective highlights the future decline of the North Water ecosystem, due to climate warming and changing sea-ice conditions, as an important climate change risk.

---

[1] Department of Glaciology and Climate, Geological Survey of Denmark and Greenland, Copenhagen, Denmark. [2] Department of Earth Sciences, University of New Brunswick, Fredericton, NB, Canada. [3] Université Laval, CNRS, UMI 3376 TAKUVIK, Québec City, QC, Canada. [4] Station Marine de Concarneau, CNRS, UMR7159 LOCEAN, Concarneau, France. [5] Department of Bioscience, Arctic Research Center, Aarhus University, Roskilde, Denmark. [6] Ecosystems and Environment Research Programme (ECRU), and Helsinki Institute of Sustainability Science, Helsinki University, Helsinki, Finland. [7] Université de Bordeaux, CNRS, UMR 5805 EPOC, Pessac, France. [8] Aarhus AMS Centre (AARAMS), Department of Physics and Astronomy, Aarhus University, Roskilde, Denmark. [9] Danish Meteorological Institute, Copenhagen, Denmark. [10] Agency for Data Supply and Efficiency, Copenhagen, Denmark. [11] Department of Geosciences and Natural Resource Management, University of Copenhagen, Københav, Denmark. [12] Department of Geoscience, Aarhus University, Aarhus, Denmark. [13] Department of Permafrost Research, Alfred Wegener Institute Helmholtz Center for Polar and Marine Research, Potsdam, Germany. [14] Department of Environmental and Biological Sciences, University of Eastern Finland, Jovensuu, Finland. [15] School of Geography, Politics and Sociology, Newcastle University, Newcastle upon Tyne, UK. [16] Department of Geography, Durham University, Durham, UK. [17] Department of Biological and Environmental Science, Nanoscience center, University of Jyväskylä, Jyväskylä, Finland. [18] Lake Group & Arctic Research Centre, Department of Bioscience, Aarhus University, Roskilde, Silkeborg, Denmark. [19] Department of Biological Sciences and Centre for Ecosystem Research and Implementation, Middle East Technical University, Ankara, Turkey. [20] Sino Danish Centre for education and Research, Beijing, China. [21] Centre for GeoGenetics, Globe Institute, University of Copenhagen, Copenhagen, Denmark. [22] Department of Biology, Limnological Institute, University of Konstanz, Konstanz, Germany. [23] Pinngortitaleriffik/ Greenland Institute for Natural Resources, Nuuk, Greenland. [24] University of Alberta, Edmonton, AB, Canada. [25] National Museum of Denmark, Copenhagen, Denmark. ✉email: sri@geus.dk; thd@bios.au.dk

The impact of changing sea-ice conditions on the productivity of resources that sustain Indigenous livelihoods in the Arctic has been identified by the IPCC as an emerging climate change risk, stemming from the triangular-intersection of an exposure, a hazard and a vulnerability[1] In this context, the well-documented millennial-scale dependence of local communities on the North Water (NOW) represents an exposure[2,3], defined by the IPCC as the presence of e.g. people, livelihoods or ecosystems in settings that could be adversely affected[1] The NOW or Pikialasorsuaq ('the great upwelling' in Greenlandic) is the largest and most productive polynya in the northern hemisphere, an annually recurring ice-free area in northern Baffin Bay (Fig. 1). The ice-free waters of the NOW allow for an enhanced and unusually early phytoplankton bloom lasting for two-four months[4] The polynya ecosystem sustains keystone Arctic species, including Arctic cod, seabirds, and marine mammals such as narwhal, beluga, walrus, and polar bear[5], which all serve to underpin the hunting and fishing economies of Inuit communities in the region[3,6] The World Conservation Union (IUCN) has identified the NOW as one of the most ecologically significant marine areas in the Arctic and proposed it as a UNESCO Natural Marine World Heritage Site due to its Outstanding Universal Value[7].

The seasonal formation and physical conditions of the NOW rely primarily on the consolidation of an ice arch or bridge across the southern Kane Basin during winter, which blocks the inflow of multiyear sea ice from the Arctic Ocean (Fig. 1). When the ice arch is stable, newly formed sea ice south of the arch is continuously removed by northerly winds and ocean currents, which

also promotes a deep mixed layer during winter and high nutrient availability for the spring phytoplankton bloom[4] Entrainment of deeper and warmer Atlantic-derived water masses further limits sea ice growth and provides nutrients that sustain high primary productivity rates[4]. Diatoms, including open-water, marginal ice zone, and sea-ice (sympagic) taxa, are the main primary producers in the NOW (e.g. ref. [8]) and although grazing pressure may vary, a significant fraction of the diatom production in the euphotic zone reaches the seafloor sediments either as intact cells and spores or empty frustules (after grazing or lysis)[9]. Following nutrient exhaustion, species belonging to the diatom genus *Chaetoceros* can produce large amounts of highly silicified, fast-sinking resting spores[10] that generally preserve well in the sediments and reflect changes in productivity levels over time.

The NOW supports >80% of the global breeding population of little auk, which is the most abundant seabird in the North Atlantic[11]. The little auk is tightly linked to the copepod *Calanus hyperboreus*, on which the chicks are raised, and >60 million birds depend on the unique availability of this prey item in the NOW[12]. By transporting vast quantities of marine-derived nutrients (MDN) from sea to land in the form of guano, little auks have transformed extensive parts of the NOW coastal landscapes into green oases[13,14]. At little auk colonies, temporal changes in the MDN flux in sediments can be used as a proxy for changes in bird numbers and, by inference, NOW productivity over time[15].

Due to its resource richness in one of Earth's most inhospitable environments, and its strategic location at the southern edge of the narrowest point between Canada and Greenland, the NOW region represents the gateway of prehistoric migrations into

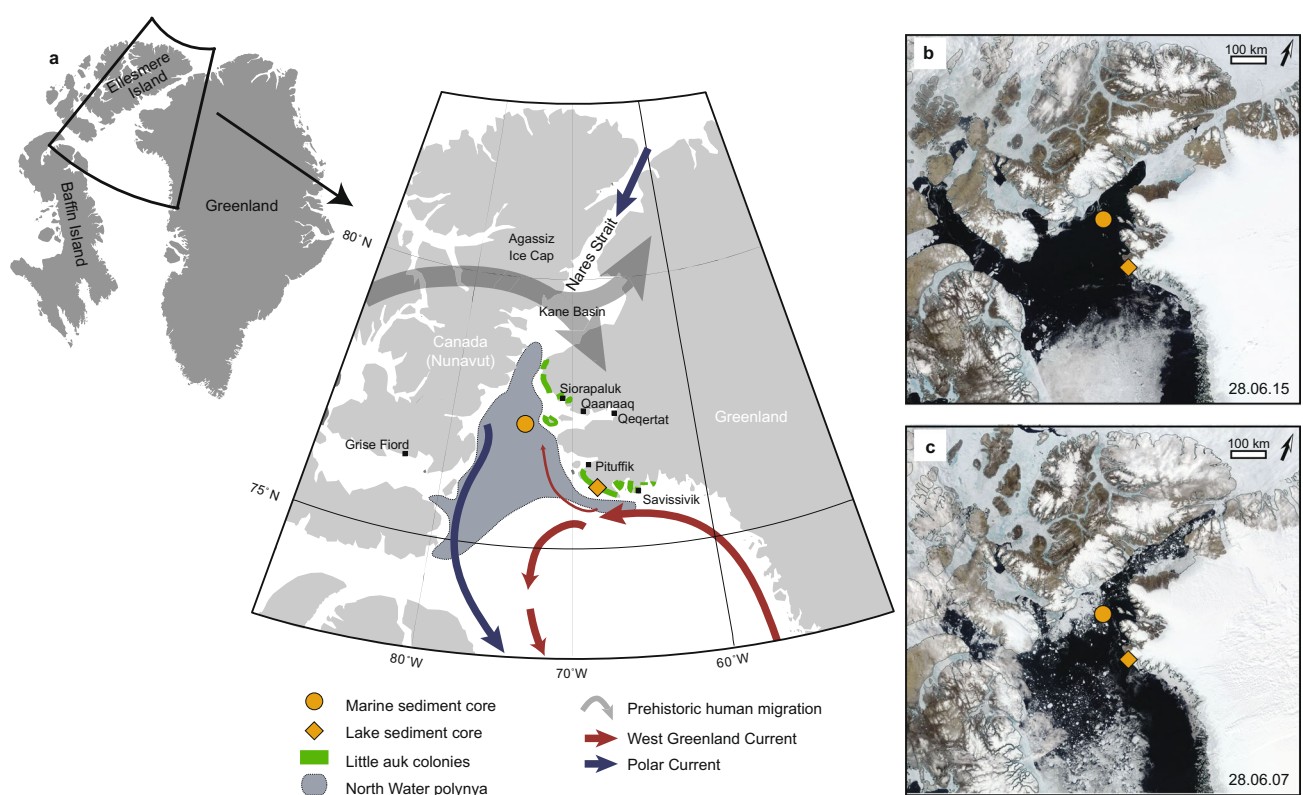

**Fig. 1 Location and configuration of the North Water polynya. a** Marine and lacustrine core sites, main ocean currents, prehistoric human migration routes, and present-day distribution of little auk colonies. The extent of the polynya is defined as in ref. [17] . **b** Example of late-June polynya configuration with a stable ice arch **c**. Example of late-June configuration in the absence of the Kane Basin ice arch. Satellite images from NASA EOSDIS. Several Inuit communities rely directly on the polynya resources today, including Qaanaaq, Siorapaluk, Qeqertat and Savissivik in Northwest Greenland, and Grise Fiord in Nunavut. Background map figures were created using Ocean Data View[73].

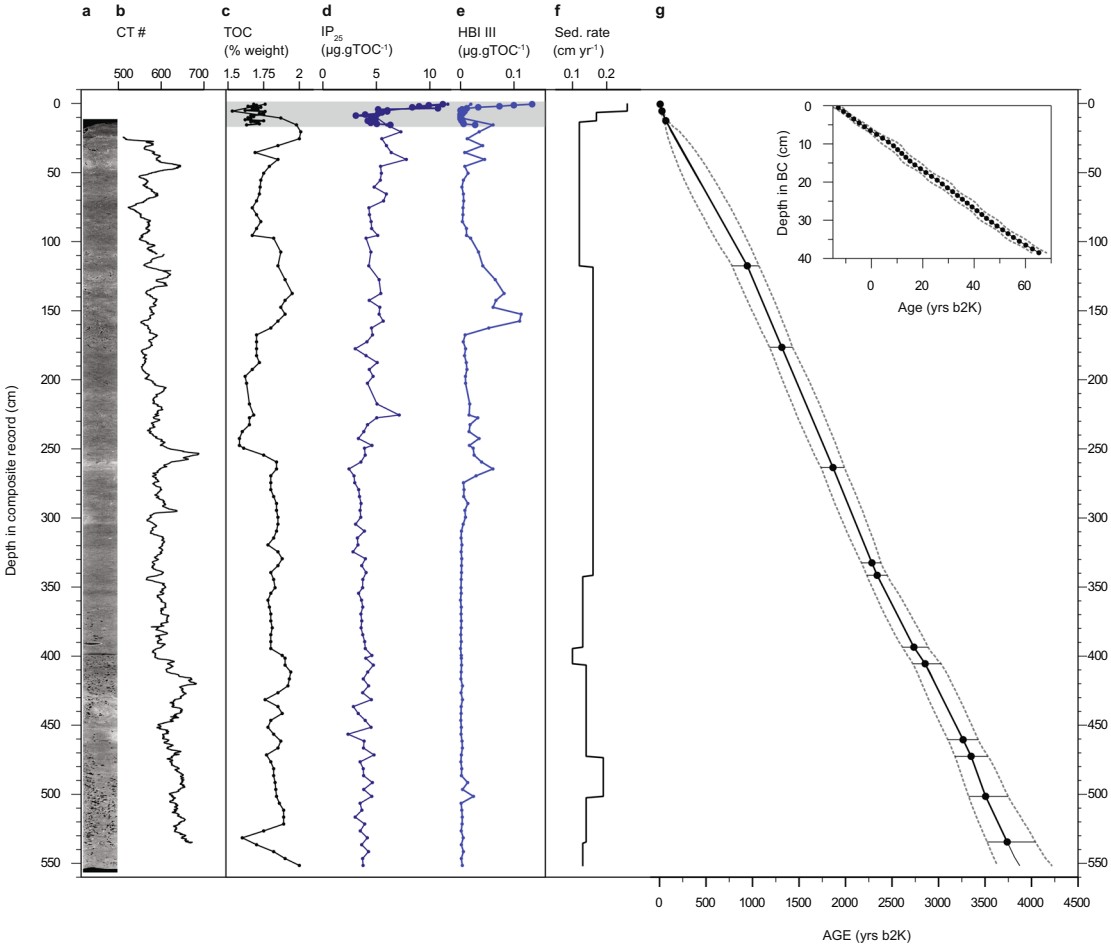

**Fig. 2 Marine sediment core. a** Computerised tomography scan image of the Casq1 core. **b** CT number. Denser areas appear whiter in the CT scan image. **c** Total organic carbon (TOC) percentage weight. **d** TOC-normalised concentrations of the sea ice biomarker $IP_{25}$. **e** TOC-normalised concentrations of HBI III (Triene). **f** Sedimentation rates. **g** Modelled median age-depth relationships constructed in BACON for CASQ1 and CASQ1 BC (insert). Error bars and dashed lines represent 95% confidence intervals (for dates see Supplementary Table 1). The grey bar indicates the stratigraphic interval covered only by the box-core record.

Greenland and has been the stage for cultural transitions since the first humans crossed the Nares Strait c. 4400 yrs b2k[2,16]. The NOW is of critical significance to Inuit communities today[6,17], and transformative shifts in the ecosystem, due to changing sea-ice conditions, represent an emerging hazard, in this context referring to a climate-related trend with negative impacts[1]. Ice-arch dynamics in Nares Strait have shown a tendency towards instability and earlier spring collapses, linked to changes in sea-ice regime and wind forcing[18–20]. This raises concerns as to whether the NOW ecosystem will persevere in a warming climate. Evaluating risks and defining climate adaptation measures for this unique ecosystem requires an understanding of past ecological and societal responses, but this knowledge is lacking due to a paucity of long-term records.

Here, we explore the third side of the NOW climate change risk triangle: vulnerability (propensity or predisposition to be adversely affected[1]). We applied a retrospective approach, reconstructing long-term trends in sea ice, primary (diatom) production and little auk colony dynamics based on two sediment core records (one marine and one lacustrine) spanning the last c. 4000 and 6000 years, respectively (Figs. 2, 3 and Supplementary Table 1). To reconstruct marine primary production, we quantified changes in the sedimentary fluxes of diatoms (diatom valves and *Chaetoceros* resting spores). As proxies of past sea-ice dynamics, we used $IP_{25}$, a source-specific molecular biomarker

produced by sympagic diatoms[21,22], and its related compound HBI III (Triene), produced by certain pelagic diatoms thriving in the cold waters of the marginal ice zone[23]. Combined, these source-specific biomarkers track polynya activity and stability over time. To infer the presence and relative abundance of little auks, we analysed $\delta^{15}N$, cadmium to titanium ratios (Cd:Ti), the fluxes of cholesterol and ß-sitosterol, and changes in diatom assemblage composition in a sediment core from a lake within the catchment of a large little auk colony (Fig. 1).

## Results and discussion
**Geochronology of sediment records and depositional environments.** The marine sediment record consists of a gravity (543 cm long) and a box core (40 cm long) retrieved from a site centrally located in Smith Sound at 692 m water depth, south of the southernmost ice arch location (Fig. 1, see 'Methods' for details). The combined record shows continuous marine sedimentation spanning the past ca. 4000 years, with sedimentation rates varying from 0.09 to 0.27 cm y$^{-1}$ in the gravity core and 0.4–0.67 cm y$^{-1}$ in the box core, and total organic carbon contents ranging 1.5–2% (Fig. 2 and Supplementary Fig. 1, Supplementary Table 1).

The lacustrine sediment core (177 cm long) was retrieved from a lake at Annikitsoq on the Cape York Peninsula at 34 m water depth and spans the last ca. 6000 years (Figs. 1, 3 and Supplementary Table 1). The lake currently lies about 1 km from the edge of the

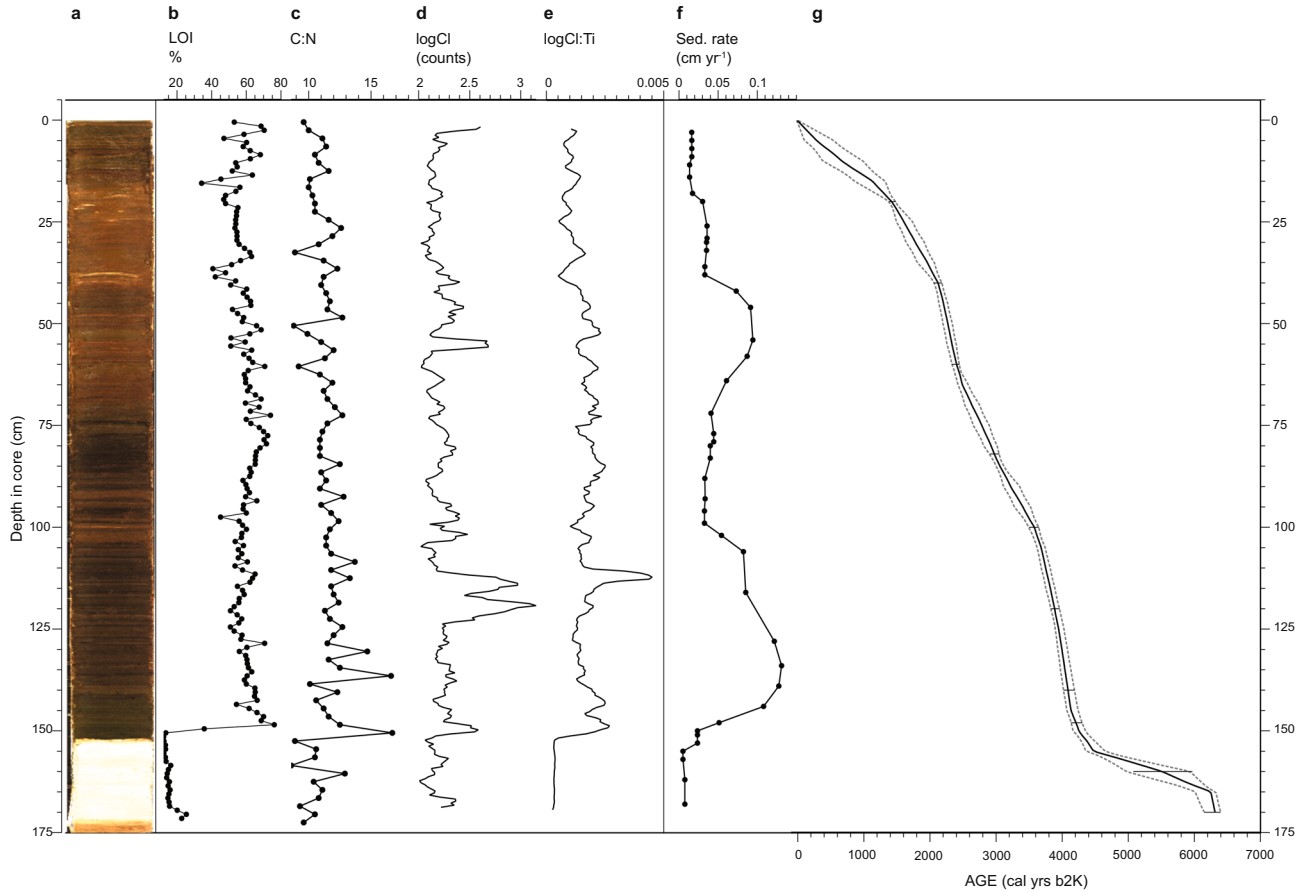

**Fig. 3 Lake sediment core. a** Core photograph showing laminations and shift from organic-poor to organic-rich sediments at c 150 cm core-depth. **b** Percentage (weight) of organic material loss on ignition (LOI). **c** Carbon to Nitrogen ratio (C:N). **d** Log of Cl counts based on X-ray fluorescence (XRF) scanning. **e** Log Cl:Ti XRF data. **f** Sedimentation rates. **g** Modelled median age-depth relationship constructed in BACON. Error bars and dashed lines represent modelled 95% confidence intervals (for dates see Supplementary Table 1).

Greenland Ice Sheet and receives some inflow from streams originating there (Supplementary Fig. 16). As such, there is an input of water with lower nutrient contents and lower $\delta^{15}N$ values than the inflow from the large little auk colony located right beside and above the lake. This lake is currently profoundly impacted by the birds with a pH of 5 and Chlorophyll-a concentration of 22.7 $\mu g\,l^{-1}$, low and high respectively for a High Arctic lake. The lake sediment core is marked by a sharp transition at 150 cm, which dates to 4400–4200 cal yrs b2k (Fig. 3). Sediment characteristics at the base of the core reflect the local geology[24], which comprises high-grade crystalline rocks (Kap York Meta-igneous Complex) with glacial silty-clays and low organic content (Loss on Ignition 20%, Fig. 3). After the transition, there is a 4-fold increase in sediment accumulation rates, the colour changes markedly, and organic contents become extremely high for a high Arctic lake (up to 80%) (Fig. 3).

Diatom analyses of the lake record, supported by XRF elemental data, provide independent evidence that the sediment record covers a period of continuous freshwater sedimentation (Fig. 3 and Supplementary Figs. 2, 3). Prior to the transition at c. 4200 cal yrs b2k, the lake diatom assemblages were dominated by Fragilarioid species (*Staurosira construens* and *Stauroforma exiguiformis*). Fragilarioids are common in Arctic lakes and ponds and are typically found in oligotrophic and circum-neutral to somewhat alkaline environments with prolonged ice cover and hence a short growing season[25]. After 4200 cal yrs b2k, there is a marked change in the lake diatom assemblages indicating a decrease in lake pH, which is seen as the dominance of

acidophilous taxa (within the genera *Psammothidium* and *Eunotia*) and an overall decrease in planktic species, which are adversely affected by lower pH[26] (Supplementary Fig. 3). The acidification of the lake is a result of the marked peat accumulation in the catchment area after the arrival of little auk (see also 'Methods'). The lake diatom assemblages show no clear signs of eutrophication despite the increased supply of marine-derived nutrients by the birds, which is likely due to the overriding effect of acidification on species composition.

**Significance of the North Water for the human settlement of Greenland**. The lake indicators record the arrival of little auks at the colony site between 4400 and 4200 cal yrs b2k (Fig. 4), corresponding to the marked transition in the core, and consistent with data from nearby terrestrial peat deposits[15]. Bird colony influence on the lake appears to be relatively stable from c. 4200 and 2700 cal yrs b2k (Figs. 4 and 5). The high diatom fluxes in the marine record indicate a productive polynya that would support bird colony expansion during this interval (Fig. 4). The high IP$_{25}$ fluxes demonstrate an active polynya with the recurrent formation of seasonal sea ice, while the minimal HBI III fluxes indicate reduced influence of marginal ice conditions at the core site (Fig. 4). Combined, the diatom and sea-ice biomarker records indicate prolonged open-water conditions consistent with an active and stable polynya. The arrival of little auk is coeval with the earliest documented human migrations into Greenland.

According to archaeological and genetic evidence, the first humans to settle in Greenland migrated from Siberia via Alaska

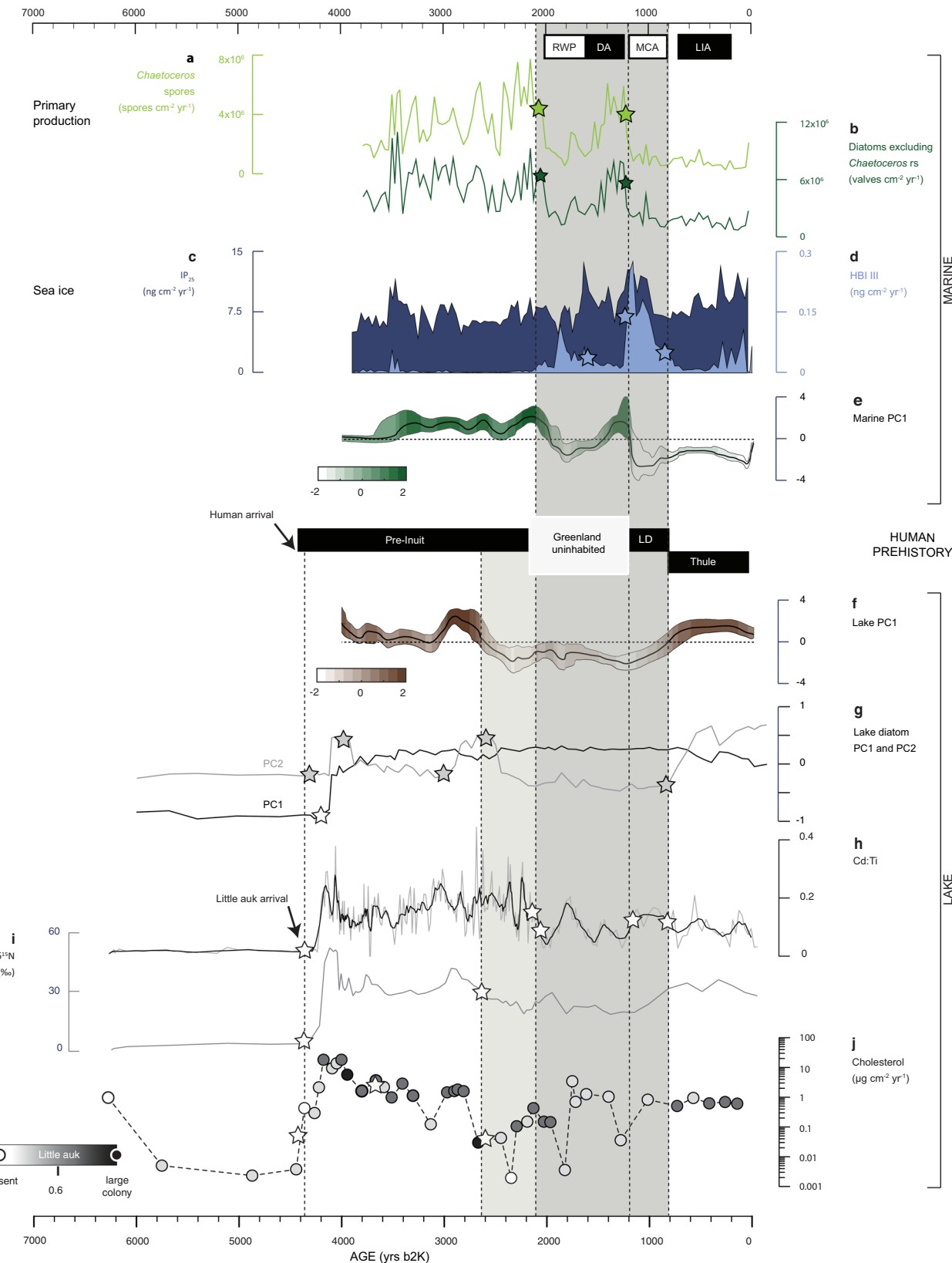

and Canada and, on arrival, followed two distinct routes, determined by the presence of the Greenland Ice Sheet[2,16]. This gave rise to the Independence I culture in Northeast Greenland[27] and the Saaqqaq culture on the West coast[28,29]—both Pre-Inuit cultures. We show that, at this time, the NOW was a stable feature that would have presented both a reliable ice bridge for

crossing from Canada to Greenland and a bountiful ecosystem (Figs. 4 and 5).

**Polynya instability and periodic human abandonment of Greenland.** During c. 2700–2200 cal yrs b2k, the lake record

**Fig. 4 Holocene changes at the North Water as evidenced by marine and lake multi-proxy records.** The blocks shaded in grey indicate periods of polynya instability. The main phases of human prehistory in Northwest Greenland are represented by bars. Stars on the individual proxy time-series indicate points of significant change based on generalised additive model (GAM) statistics (see 'Methods' and Supplementary Figs. 7–15), and dashed vertical lines denote significant changes in more than one proxy at the same time. **a** Marine primary production as indicated by the fluxes of *Chaetoceros* resting spores. **b** Marine primary production as indicated by fluxes of diatoms (excluding *Chaetoceros* resting spores). **c** Fluxes of the sea ice biomarker IP$_{25}$. **d** Fluxes of the ice-marginal zone indicator HBI III (z triene). **e** Principal component 1 for the marine record proxies (Supplementary Fig. 4 and Supplementary Table 2). **f** Principal component 1 for the lake record proxies (Supplementary Figs. 5, 6 and Supplementary Table 2). **g** Principal Components 1 and 2 for the lake diatom assemblages (Supplementary Fig. 3). **h** Changes in Cadmium (Cd) to Titanium (Ti) ratios tracing the level of Cd-enrichment of the lake sediments by bird guano. The dark line shows a LOESS smoothed curve. **i** δ$^{15}$. **j** A combination of total cholesterol flux, and the ratio of cholesterol to cholesterol plus β-sitosterol (greyscale) indicates relative bird colony size. RWP Roman Warm Period, DA Dark Ages cold period, MCA Medieval Climate Anomaly, LIA Little Ice Age, LD Late Dorset.

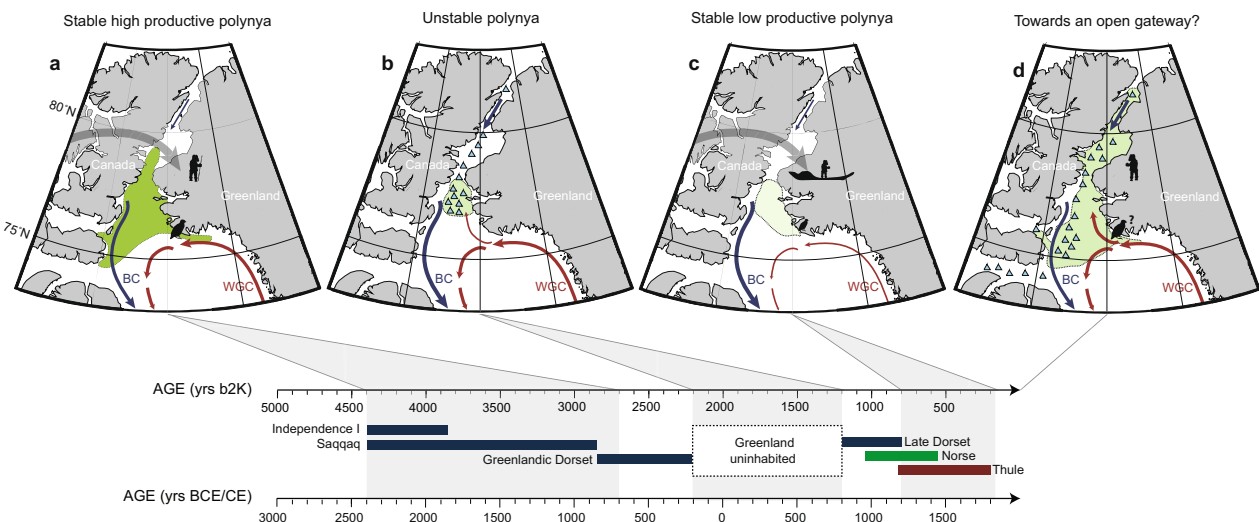

**Fig. 5 Evolution of the North Water ecosystem and cultural transitions in Greenland. a** A stable and highly productive polynya is inferred from our records after 4400–4200 cal yrs b2k, coincident with the arrival of the first humans in Greenland and the first appearance and expansion of little auks in the area. **b** From 2700 to 800 cal yrs b2k, the polynya is unstable and reduced in extent, particularly after 2200 cal yrs b2k. This period spans a void in the human settlement of Greenland from **c**. 2200–1200 yrs b2k and absence/low abundance of little auks. **c** From c. 800 cal yrs b2k, a stable but low productive polynya is inferred and little auk colonies recover. During this time, there is a replacement of Late Dorset groups by the Thule Culture, the direct ancestors of modern Inuit. **d** Predicted disappearance of the polynya following the current trajectory of Arctic warming and sea-ice decline. BC Baffin Current, WGC West Greenland Current. Changes in WGC influence in the polynya region are based on ref. [49]. Triangles represent drift ice. Shades of green represent the interpreted late-spring relative extent and productivity of the polynya (darker green corresponds to a more productive polynya and vice-versa). Background map figures were created using Ocean Data View[73].

indicates a significant decline in the abundance of little auks, culminating in a short-lived abandonment of the colony c. 2300 cal yrs b2k (Fig. 4f–i). This decline is marked by falling δ$^{15}$N and cholesterol values, and a reduction of moderately to strongly acidophilous diatom species (*Psammothidium marginulatum*, *Psammothidium helveticum*, and *Eunotia* species)[30,31] (Supplementary Fig. 3). In the marine record, primary production remains high until c. 2200 cal yrs b2k after when it drops significantly (Fig. 4). The fact that the lake record indicators show an earlier decline than the marine record indicators, is presumably linked to the different locations of coring sites relative to the spatial extent of the polynya. The marine coring site lies at the centre of the polynya, whereas the lake site is located close to its present-day southern edge, and therefore any polynya contractions would be recorded somewhat earlier at this location (Fig. 1). Although dating uncertainties cannot be ruled out as an alternative explanation, this idea is supported by peat core studies from further north, demonstrating that bird colonies spread northwards around 2800 and 2200 cal yrs b2K[15].

Both records indicate a sustained period of polynya contraction and instability between 2700/2200–800 cal yrs b2k (Fig. 4). This period encompasses a long-term void in the

human prehistory of Greenland, spanning both the Roman Warm Period and the Dark Ages Cold Period (Figs. 4 and 5). Archaeological studies point to Greenland being uninhabited for about a millennium from the disappearance of the Greenland Dorset in West Greenland at c. 2200 cal yrs b2k until the arrival of the late Dorset Culture by c. 1200 cal yrs b2k[32] (Fig. 5). The Late Dorset inhabited Northwest Greenland during the Medieval Climate Anomaly[33] (Figs. 4 and 5).

Between c. 800 and 100 cal yrs b2k, atmospheric cooling intensified, reaching temperatures up to 3 °C degrees cooler than at the time of human and bird arrival c. 4400–4200 cal yrs b2k, consistent with cooler sea-surface temperatures in the North Atlantic region during the Little Ice Age (refs.[34,35]; Fig. 6). A return to more stable conditions is suggested by the sea-ice biomarkers, and while marine diatom fluxes remain relatively low (Fig. 4), the little auk colony appears to recover after c. 800 cal yrs b2k (Fig. 4), reflected by rising δ$^{15}$N and a higher ratio of cholesterol to cholesterol plus β-sitosterol, along with a return to lake diatom assemblages dominated by acidophilous species (Supplementary Fig. 3). This period is marked by another significant cultural transition: the arrival and rapid expansion of the Thule Culture, the direct ancestors of modern Inuit (refs.[16,34]; Fig. 5).

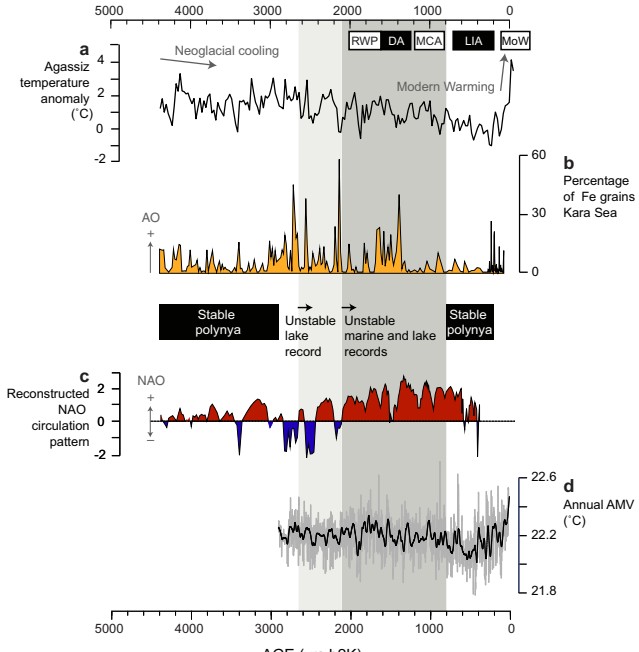

**Fig. 6 Climate variability in the Arctic and North Atlantic region during the late Holocene from selected high-resolution records. a** Air temperature anomalies from the Agassiz ice core record from[52]. **b** Arctic Oscillation (AO) reconstruction from a Kara Sea record as presented in ref. [45]. **c** North Atlantic Oscillation (NAO) reconstruction from a southwest Greenland lake record as presented in ref. [44]. **d** Atlantic multidecadal variability (AMV) as reconstructed from a lake record from Ellesmere Island[39]. RWP Roman Warm Period, DA Dark Ages Cold Period, MCA Medieval Climate Anomaly, LIA Little Ice Age, MoW Modern Warming. Grey bars indicate periods of polynya instability as inferred by our lake and marine records. The darker grey bar overlaps with the period of human abandonment of Greenland from c. 2200–1200 cal yrs b2k.

**Climate forcing of polynya dynamics.** Polynya dynamics in the North Water are strongly linked to sea ice conditions (sea ice thickness, concentration and motion) and prevailing winds, and thus influenced by local and regional climate forcing. The inception of recurrent ice arches in the Nares Strait region is estimated to have occurred at the mid-to-late Holocene transition, following atmospheric cooling and extensive sea ice formation in the Canadian Arctic and Greenland[35,36]. The North Atlantic Oscillation (NAO) and the Arctic Oscillation (AO) are the two dominant (and closely related) modes of atmospheric variability for the mid-latitudes of the North Atlantic region and the entire Northern Hemisphere, respectively[37,38]. The NAO/AO have shown periodicities on multidecadal timescales that are linked with sea-surface temperature changes in the North Atlantic (Atlantic Multidecadal Variability (AMV)), in turn influencing Arctic sea-ice variability[39,40]. Sea ice concentration and motion in the Arctic Ocean region are also highly impacted by the Dipole Anomaly (DA) pattern—the second mode of winter variability - which promotes sea ice transport from the western to the eastern Arctic Ocean and export towards the North Atlantic through Fram Strait, and appears to be particularly important during warmer periods[41–43].

From c. 4400 to 2700 cal yrs b2k, the polynya remained relatively stable and productive, and paleo-records point towards predominantly weakly positive NAO and low magnitude variations in the AO during this period (refs.[44,45], Fig. 6). In contrast, the period of polynya instability c. 2700/2200–800 cal yrs b2k encompasses episodes of abrupt climate anomalies in the

North Atlantic region, characterised by enhanced Arctic water influence in the subpolar North Atlantic[46]. Ocean-atmosphere climate shifts after 2700 yrs b2k have been linked to a strengthening of the latitudinal temperature gradient and southward migration of the Polar Front[46,47], coincident with cooling and freshening of bottom waters off the West Greenland coast from 2600 to 1900 cal yrs b2k[48] The period of polynya instability spans two warm(er) intervals, the Roman Warm Period and the Medieval Climate Anomaly. Marine records from offshore Greenland indicate ocean warming, particularly during the Roman Warm Period (c. 2000–1600 cal yrs b2k), linked to an increased contribution of Atlantic-sourced waters to the West Greenland Current[48,49].

Predominantly positive modes of the AO (and NAO) are inferred for the period of human abandonment of Greenland c. 2200–1200 cal yrs b2k and coeval with our demonstrated decline in NOW productivity (Figs. 4 and 6). Several episodes of strong positive Arctic Oscillation (AO) anomalies of magnitudes unprecedented in the Holocene are indicated by Kara Sea-sourced ice-rafted iron grains preserved in sediment cores from the Alaskan coast (ref.[45], Fig. 6). Positive winter AO conditions lead to cyclonic wind activity promoting sea ice motion and export out of the Arctic Ocean[50]. Such conditions can have a negative impact on ice arch formation and stability, and lead to an increase in sea ice drift in the NOW region, as demonstrated by satellite data (Fig. 7). While the AO affects sea ice conditions in the NOW, the Dipole Anomaly has been determinant in promoting sea ice loss from the Arctic Ocean via the Transpolar Drift system in recent decades[41]. This partly explains why negative AO excursions in recent years have also led to record low Arctic sea ice volumes, contributing to amplifying ice loss and warming in the region (ref.[51], Fig. 7).

**The future of the North Water in a warming climate.** Our long-term perspective on the evolution of the NOW underscores the tight coupling of this ecosystem to climate forcing and highlights its vulnerability to climate change. The concordance of past periods of decreased marine productivity with human abandonment of Greenland suggests that the future collapse of the NOW ecosystem is a legitimate climate change risk.

Multiple lines of evidence suggest increasing polynya instability over the last two decades. Remote sensing data indicate earlier ice arch break-up, leading to sea ice export from the Arctic Ocean into the Baffin Bay via the Nares Strait[18,19], which allows for increased late spring/summer ice drift and melt in the NOW region as ice is not held upstream of the arch (Fig. 7). This is reflected in our marine record as increased fluxes of both sea ice biomarkers (Fig. 7). Present-day air temperatures in the High Arctic are unprecedented in the history of the NOW[52], and anthropogenic warming is expected to exacerbate Arctic sea ice thinning and loss, detrimental to ice arch stability and polynya formation.

On our present climate trajectory, the NOW will likely cease to exist as a globally unique ice-bounded open-water ecosystem, and a winter refuge for keystone High Arctic species[53]. The vulnerability of the NOW is a clear example of the emerging climate change risk associated with changing sea ice conditions on the productivity of indigenously harvested resources[1]. The unprecedented speed of the changes already underway should provide an impetus to ensure that climate change does not outpace the adaptive capacity and threaten the anticipatory knowledge of High Arctic Inuit communities, who are also facing social and economic challenges. This is a priority highlighted by the Inuit Circumpolar Council's Pikialasorsuaq Commission, which recommends that the governments of

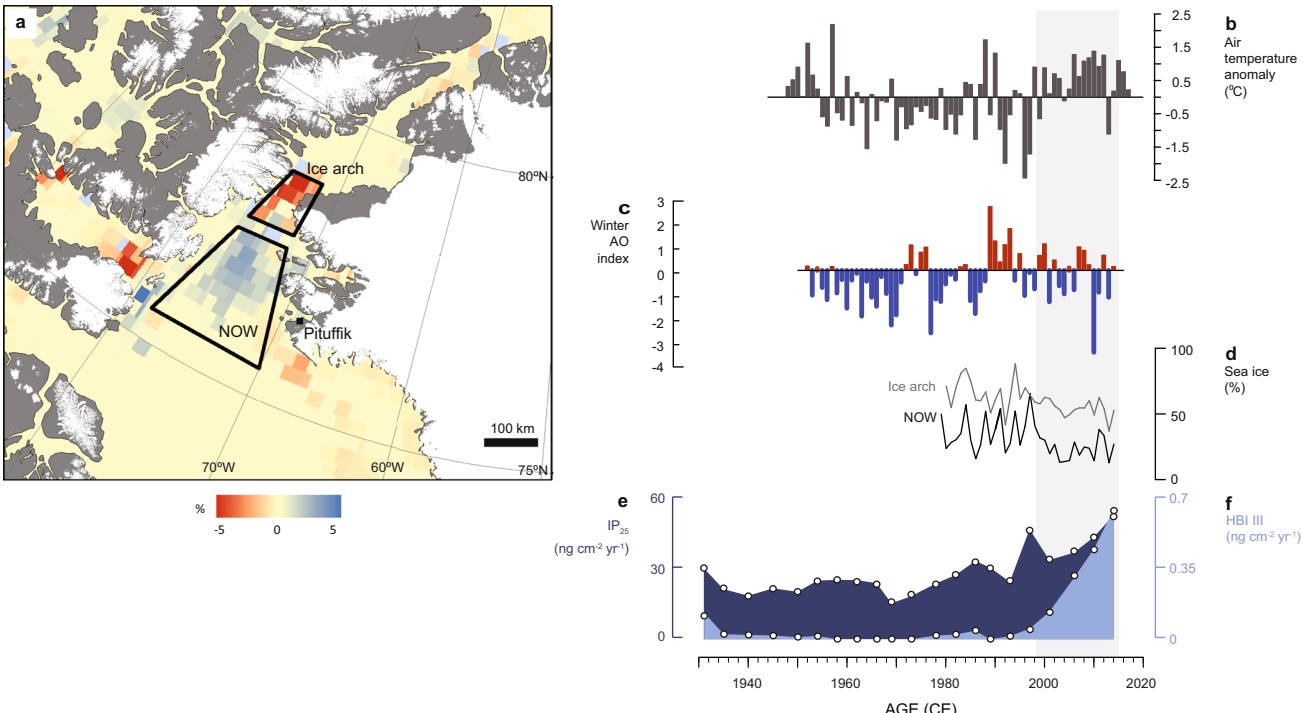

**Fig. 7 Trends in atmospheric and sea ice conditions in the North Water region since the mid-twentieth century. a** Anomaly map showing the difference in May–September sea ice concentration following the extreme AO + anomaly year of 1989 compared to the extreme AO-anomaly year of 1996 and location of the weather station at Pituffik (Thule Air Base). Extreme AO + conditions resulted in 5% sea-ice cover reduction in the ice arch area and an increase in drift ice in the polynya area compared to AO- conditions. **b** Air temperature anomaly for May–September at Pituffik. **c** Winter AO index. **d**. Average sea ice concentrations during May–September for the two regions of interest based on satellite observations (see Methods for details); **e, f** Fluxes of the sea ice biomarkers $IP_{25}$ and HBI III, respectively, in the box-core marine sediment record. The grey bar denotes a regime shift in sea ice (declining %) and air temperatures (positive anomalies) after 1998.

Greenland and Canada support ecosystem monitoring and conservation of living resources, and work with indigenous organisations and local communities to implement a management regime for the polynya, and the creation of an Indigenous Protected Area (IPA)[17].

## Methods

**Marine sediment record**. The Calypso Square gravity core AMD15-CASQ1 (77° 15.035′ N, 74°25.500′ W, 692 m water depth) and accompanying box core (BC; same location) were retrieved aboard the CCGS *Amundsen* during the ArcticNet 2015 Leg 4a expedition in 2015, in accordance with relevant permits and local laws. The CASQ corer recovered a sequence 543 cm long, while the box core was 40 cm long. Sediment material from these cores is stored at the Geological Survey of Denmark and Greenland and available upon reasonable request to the first and corresponding author (SRI).

Computed Tomography (CT) scanning of the core was performed using a Siemens SOMATOM Definition AS + 128 at the Institut National de la Recherche Scientifique (INRS), Quebec, Canada. The tomograms were converted into digital DICOM format using a standard Hounsfield scale (HU scale) from −1024 to 3071, where −1024 corresponds to the density of air, 0 to the density of water and 2500 to the density of calcite.

The age control on the marine sediment record was provided by 11 accelerator mass spectrometry (AMS) radiocarbon dates on mollusc shells (Supplementary. Table 1) at the Keck Carbon Cycle AMS Facility, University of California, Irvine, US, and $^{210}Pb/^{137}Cs$ measurements conducted on 20 samples at the Gamma Dating Center, Copenhagen University, Denmark. In the box core, the content of unsupported $^{210}Pb$ showed a clear exponential decline with depth (Supplementary Fig. 1). A clear $^{137}Cs$ peak was not detected, but the $^{210}Pb$-based chronology dates the earliest sample with $^{137}Cs$ to 1969 ± 2 years, which is close to the expected date, 1963, for the global $^{137}Cs$ peak induced by nuclear weapons testing in the atmosphere. This, and the very uniform exponential decline in unsupported $^{210}Pb$ with depth, gives confidence in the calculated chronology. A mixed age-depth model, using both $^{210}Pb$ and $^{14}C$ dates, was constructed using BACON, an open-source package of 'R'[54]. This Bayesian accumulation model code allows for greater flexibility in sedimentation rates between dated intervals than traditional linear age-depth models[54]. The AMS radiocarbon dates were calibrated with the Marine13 IntCal13[55], and the regional marine reservoir offset was estimated based

on existing $^{14}C$ data from marine specimens collected before the mid-1950s. Distinct regional offset values have been proposed for Arctic Canada, but do not include the Smith Sound region[56]. Existing data from NW Greenland show local reservoir correction (ΔR) values ranging from -40 years in the Inglefield Fjord to +320 years in Ellesmere Island (the latter consistent with the proposed 335 ± 85 years for the Canadian Arctic Archipelago[56]). However, these samples have been retrieved from shallow sites (<40 m water depth), which are unlikely to reflect influence from the West Greenland Current. Data from the only deeper site in the NOW region are based on measurements of the mollusc *Astarte montagui* and indicate an offset of 140 ± 60 years[57]. This value is consistent with measurements from the Disko Bay region, also under the influence of the West Greenland current[58]. We have therefore chosen to use an offset value of 140 ± 60 years. Ages are reported in calibrated years before 2000 (cal yrs b2k) and year CE (for the box-core record presented in Fig. 7).

TOC measurements were carried out at the Geological Survey of Denmark and Greenland, at 5 cm intervals in the CASQ core and at 2 cm intervals in the box-core. Dried sediment samples (~0.5 g) were powdered (<250 micron) and subjected to Rock-Eval type bulk flow pyrolysis using a HAWK instrument (Wildcat Technologies, Texas). Sets of one control-standard (in-house standard) and one blank were run every 10 samples to ensure instrument stability.

**Sea ice biomarkers**. A small number of common pan-Arctic diatoms belonging to the *Haslea* and *Pleurosigma* genera are known to produce the sea ice biomarker $IP_{25}$—a mono-unsaturated highly branched isoprenoid (HBI) alkene, biosynthesized in the sea-ice matrix[21,22] and deposited in marine sediments following ice melt. Due to its source-specificity and good preservation potential in marine sediments, $IP_{25}$ constitutes direct evidence for seasonal sea ice. In marine settings, high sedimentary $IP_{25}$ content generally reflects increasing seasonal sea ice concentrations, whereas the absence of $IP_{25}$ can either indicate perennial sea-ice cover or open-water[23]. As such, downcore changes in sedimentary $IP_{25}$ fluxes from a given location can be interpreted to reflect temporal fluctuations in sea ice conditions. A related lipid biomarker HBI III is produced by diatoms blooming in the often ice-loaded and relatively fresher and cooler surface waters typical of the marginal ice zone (e.g. ref.[59]). It has recently been shown that the relative abundances of $IP_{25}$ and other HBIs remain essentially unaltered in trophic food webs and faecal pellets[60]. These findings suggest that source HBI distributions remain unaltered following grazing, which implies that changes in grazing efficiency do not have a significant impact on the sedimentary signature of HBIs.

Prior to analytical treatment, an internal standard (7-hexylnonadecane) was added to 0.5 g of freeze-dried and homogenised sediment. Total lipids were ultrasonically extracted (×3) using a mixture of dichloromethane (DCM: $CH_2Cl_2$) and methanol (MeOH) (2:1, v/v). Extracts were pooled together, and the solvent was removed by evaporation under a slow stream of nitrogen. The total extract was subsequently resuspended in hexane and purified through open column chromatography ($SiO_2$). Hydrocarbons (including $IP_{25}$ and HBI III) were eluted using hexane (8 mL). Procedural blanks and standard sediments were analysed every 15 samples. Hydrocarbon fractions were analysed using an Agilent 7890 gas chromatograph (GC) fitted with 30 m fused silica Agilent J&C GC columns (0.25 mm i.d. and 0.25 μm phase thickness) and coupled to an Agilent 5975 C Series Mass Selective Detector (MSD). The following oven temperature programme was used: 40–300 °C at 10 °C min$^{-1}$, followed by an isothermal interval at 300 °C for 10 min. The data were collected using Chemstation and analysed using the MassHunter quantification software. $IP_{25}$ and HBI III were identified on the basis of retention time and comparison of mass spectra with authenticated standards. Abundances were obtained by comparison between individual GC–MS responses relative to those of the internal standard. Biomarker data presented here are reported as fluxes to account for changes in sedimentation rate.

**Changes in marine primary production**. Diatom fluxes were used to infer changes in marine primary production. For diatom quantification, sediment samples were treated with hydrogen peroxide ($H_2O_2$, 30%) and hydrochloric acid (HCl, 10%) to remove the organic material and carbonate, respectively. Residues were then rinsed several times with distilled water. A known volume of the final residue, homogenised in suspension, was added to a coverslip. Once the samples were completely dried, microscopy slides were mounted in Naphrax® for observation. Quantification of diatom valves was done using an optical microscope (Olympus BX43) with phase contrast optics at a magnification of 1000x. Concentrations were calculated based on the surface area of the slide that was analysed. Fluxes were calculated by combining diatom concentrations (ind. g$^{-1}$) with mass accumulation rates (g cm$^{-2}$ yr$^{-1}$).

**Lake sediment record**. Sediment core NOW25c was collected from a lake at Annikitsoq on the Cape York Peninsula (76°2.100′ N, 67°36.540 W, 8.1 m a.s.l.) on July 30th 2015. The sediment core was recovered from 34 m water depth using a highly portable piston corer specially adapted for remote location use. The 177cm-long core was kept upright and drained of water, the sediment surface was secured by packing the core top with a rigid foam block (Oasis), and the core was kept cool and dark before and during transport from Greenland to Denmark. Fieldwork was conducted in accordance with local laws and permits. Sediment material is deposited at Aarhus University and available upon reasonable request from the last author (TAD).

The lake was not stratified at the time of sampling (July 30–August 2 2015) and was partially ice-covered on the day of arrival. The surface waters were 4 °C and oxygen saturation was over 100% all the way to the lakebed.

The age control of NOW25c was attained by 10 accelerator mass spectrometry (AMS) radiocarbon dates at Aarhus AMS Centre (AARAMS), Aarhus University, 9 on terrestrial moss remains and 1 on humic extraction of a bulk sample (Supplementary Table 1). The radiocarbon ages of the samples were converted into calendar years using the IntCal13 calibration curve[55]. The age model was calculated using the R routine BACON[54]. Ages reported and used in the figures are median modelled ages converted to calibrated years before 2000 (cal yrs b2k).

The Loss on Ignition (LOI) technique[61] was used to determine the organic matter content at a 1 cm resolution. The sediment was dried to calculate water content and then heated to 550 °C for two hours and reweighed to calculate the percentage organic matter.

The lake sediment core was split along its length then placed in an ITRAX core scanner to obtain high-resolution pictures and measure micro-XRF. The XRF scans were made at the Aarhus University core scanning facility with a molybdenum tube set at 30 kV and 30 mA with a dwell time of 4 s. Prior to analysis, the sediment surface was flattened and covered with a 4 μm ultralene film. A step size of 0.1 mm was selected to capture possible elemental variations even in small laminations. Count readings for less abundant elements, such as Ti, maybe too low with a 4 s dwell time, so counts were summed to at least 1 mm for analysis and presentation.

**Tracking the presence, absence, and relative abundance of little auks**. We used a combination of δ15N (stable isotope of nitrogen), the ratio of Cadmium (Cd) and Titanium (Ti) concentrations in the lake sediments, concentrations and fluxes of cholesterol and β-sitosterol and diatom assemblage composition changes to assess the presence/absence and relative size of the adjacent little auk colony through time. δ15N differs markedly between marine and freshwater systems and has been shown to provide an unequivocal signal of marine-derived nutrients (MDN)[13]. Cd is also more concentrated in the marine system and thus can be used to trace seabird influence[62]. Cd is abundant in the seabird excrement, but it may also be present in the lake's catchment and so the ratio Cd:Ti is used, as Ti is a proxy of catchment input to the lake. The development of soil, peat and permafrost in the catchment stores C, N and perhaps to a lesser extent Cd, reducing the quantities delivered to the lake and affecting isotope fractionation. The extremely

high values of δ15N and cholesterol at the time of bird arrival likely reflect the absence of soil in the catchment and the unimpeded flow of bird-derived compounds into the lake. In the event that the inputs of C, N and Cd are reduced due to reduced supply as bird numbers fall, the catchment has the potential to act as a source. Thus, whilst the isotopes of N give a clear signal of bird arrival, they have the potential to be less reliable in tracking absence as a result of their storage and subsequent release. This may depend on the residence time of the lake and the degree of flushing by non-bird impacted waters. The combination of total cholesterol flux and the ratio of cholesterol to cholesterol plus β-sitosterol has been used to identify marine bird influence in freshwater systems in the Arctic (e.g. ref. [63]).

Marine zooplankton, upon which the little auks feed, are rich in cholesterol but contain virtually no β-sitosterol. In contrast, terrestrial and freshwater primary producers contain a high proportion of β-sitosterol[64]. We found very high cholesterol concentrations (1497 μg g$^{-1}$) and low β-sitosterol concentrations (15 μg g$^{-1}$) in little auk excrement, contrasting with goose excrement collected in the same region (16 μg g$^{-1}$ of cholesterol and 159 μg g$^{-1}$ of β-sitosterol), a species which largely grazes on terrestrial vegetation. Thus, when seabirds are abundant, cholesterol values in the lake sediments can be expected to be high, and the ratio of cholesterol to cholesterol plus β-sitosterol should also be high. In the NOW region, the ratio is 10-fold higher in little auk excrement (0.99) compared with goose excrement (0.09). This ratio of cholesterol to cholesterol plus β-sitosterol has been previously used as an index of seabird influence, with a value of around 0.6 indicating that the majority of the cholesterol originates from seabirds[64]. This index was derived from studies of fulmar colonies in relatively oligotrophic systems, which is not the case at Annikitsoq, where the lake is situated in extremely eutrophic systems (due to the little auk colony) with lush vegetation in the catchment and freshwater algae thriving in the water column, both of which are sources of β-sitosterol. Here, the arrival of little auks, and with them large quantities of MDN, transformed nutrient availability, prompting a period of exceptionally high terrestrial and aquatic biological productivity. This is demonstrated by the fact that 2 m of peat accumulated in the catchment adjacent to the lake in just 1000 years, following little auk colonisation[65], and the extremely high LOI values recorded in the lake (75% instead of <10% as expected for a High Arctic lake). Therefore, in addition to the large input of marine-derived cholesterol, there was also the input of cholesterol of terrestrial and freshwater origin, which has a higher proportion of β-sitosterol. Thus, a slight drop in bird input combined with increased terrestrial production, as nutrient levels remain sufficient, would result in a lower ratio, even when birds may still be present. Thus, whilst the ratio is still a useful indicator (especially when <0.4), we have not used a cut-off value signalling the dominance of seabird influence but instead simply present the index value over time. Bird absence is most likely when both cholesterol and the ratio of cholesterol to cholesterol plus β-sitosterol are low.

**Sterol analysis**. Sterol analysis followed standard protocols[66]. Specifically, Androstanol (0.1 mg mL$^{-1}$) was added as an internal standard to each sample of approximately 0.5 g of dried, homogenised sediment. Lipid compounds were extracted with solvents (DCM:MeOH, 3:1) using Microwave-Assisted Extraction, saponified and separated into neutral and acid fractions using aminopropyl SPE columns. The neutral fraction of each sample was then separated using silica gel column chromatography. The sterol fraction was trimethylsilylated using N,O-*bis* (trimethylsilyl)trifluoroacetamide (BSTFA)/trimethylchlorosilane (TMCS) (99:1 v/ v) and heated at 70 °C overnight. Excess BSTFA-TMCS was removed by drying gently under nitrogen. Samples were dissolved in 50–100 μl of ethyl acetate prior to gas chromatography-flame ionisation detection (GC-FID) and gas chromatography–mass spectrometry (GC–MS) analysis. GC–MS analyses were performed on an Agilent 7890B GC injector (280 °C) linked to an Agilent 5977B MSD in full scan mode (50–600 amu s$^{-1}$). Separation was performed on an Agilent fused silica capillary column (HP-5, 60 m, 0.25 mm ID, 0.25 um df) with Helium as a carrier gas. Sterol derivatives were analysed using the following temperature programme: 50 °C (held for 2 min) to 200 °C at 10 °C min$^{-1}$, then to 300 °C at 4 °C min$^{-1}$ and held for 20 min. GC–MS peaks were identified through comparisons with known mass spectra (NIST08) and standards where possible. Analytes were quantified based on internal standards. For 13 of the samples, lipids were extracted from the dry sediment using a chloroform:methanol 2:1 mixture and then soni-cated for 10 min, after which 0.75 mL of distilled water was added. Lipids were fractionated into neutral lipids (NLs; including sterols), glycolipids, and phospholipids (PLs) using a Bond Elut (0.5 mg) silica cartridge. First, the resin of the cartridges was conditioned using 5 mL of chloroform. Subsequently, the total lipids (1 mL) were applied to the resin, rinsed using chloroform, and then the NLs (including sterols) were collected under vacuum using 10 mL of chloroform. Sterols from the NL fraction were silylated with BSTFA, TMCS, and pyridine at 70 °C for 1 h. Trimethylsilyl (TMS) derivatives of sterols were analysed with GC–MS (Shimadzu) and GC-FID (Shimadzu) equipped with a Phenomenex (USA) ZB-5 Guardian column (30 m × 0.25 mm × 0.25 μm). Cholesterol and β-sitosterol were identified using characteristic ions of GC–MS runs[64] and quantified with GC-FID using authentic standard solutions of plant sterol mixture from Larodan (including 53% β-sitosterol, 7% stigmasterol, 26% campesterol, 13% brassicasterol), and cholesterol from Sigma-Aldrich. The recovery percentage of the sterol samples was calculated using 5-α-cholestane (Sigma-Aldrich) as an internal standard.

**Stable isotope analyses**. For stable isotope analysis, samples were taken at 2 cm intervals, freeze-dried for 48 h and ground into fine powder. The samples were packed into tin cups and analysed at UC Davis Stable Isotope Facilities, California, USA. Here, carbon ($^{13}$C) and nitrogen ($^{15}$N) isotope analyses were conducted using an elemental analyser and a continuous flow isotope ratio mass spectrometer (IRMS). Specifically, an Elementar Vario EL Cube (Elementar Analysensysteme GmbH, Hanau, Germany) interfaced with an Isoprime VisION IRMS (Elementar UK Ltd, Cheadle, UK). Samples were combusted at 1080 °C in a reactor packed with chromium oxide and silvered copper oxide. After combustion, a reduction reactor trap removed oxides and a helium carrier then flowed through a water trap (magnesium perchlorate and phosphorous pentoxide). $CO_2$ was held in an adsorption trap until the $N_2$ peak was analysed; after which the $CO_2$ was released by heating to the IRMS. Reference materials included: IAEA-600, USGS-40, USGS-41, USGS-42, USGS-43, USGS-61, USGS-64, and USGS-65. A sample's isotope ratio is expressed relative to a reference gas peak analysed with each sample. These provisional values are finalised by correcting the values for the entire batch based on the known values of the included laboratory reference materials. The long-term standard deviation was 0.2 per mil for $^{13}$C and 0.3 per mil for $^{15}$N. The delta values are expressed relative to international standards VPDB (Vienna Pee Dee Belemnite) and Air for carbon and nitrogen, respectively.

**Diatom analyses of the lake record**. Diatom analyses were carried out on 77 samples at a resolution of 1–3 cm, covering the entire lake record. Sediment samples were cleaned using $H_2O_2$ and HCl, and permanent slides were prepared using Naphrax®. A minimum number of 400 valves were counted per sample, and relative species abundances were calculated as percentages of the total counts in each sample.

**Modern sea ice concentration in the NOW region**. We assessed mean sea-ice concentration during the months May–September (MJJAS) in two regions of interest—NOW and Ice Arch (Fig. 7). We used the gridded sea ice concentration and extent product based on satellite observations during 1979–2015, available from the NOAA/NSIDC Climate Data Record of Passive Microwave Sea Ice Concentration[67].

**Numerical methods**

*Monte Carlo simulations and principal components analysis*. To derive principal components of the marine and lake records, we first performed Monte Carlo simulations of the constituent time series from each record (Supplementary Figs. 4–6). We performed 10,000 simulations of both the marine and lake records that acknowledged both measurement and dating uncertainties of the constituent time series. The 10,000 Monte Carlo simulations yield 2D frequency histograms depicting the probability density of a given measurement at a given time for each time series. We took the median (50th percentile) as the 'most likely' time series for each variable and used the 5th and 95th percentiles to define a 90% confidence envelope around this 'most likely' time series. For the marine record, each of the three constituent time series analysed (Diatom flux, *Chaetoceros* spores flux, and HBI III) were assumed to have a measurement uncertainty of ±7.5% (5–10%) and the depth-dependent dating uncertainty shown in Fig. 2, assumed to represent 2-sigma uncertainty. For the lake record, the four constituent time series were assumed to have measurement uncertainties of 2% for Cd:Ti, 0.5 per mil for $\delta^{15}$N, and 0.001 mg cm$^{-2}$ yr for fractional sterol, with depth-dependent dating uncertainty (as shown in Fig. 3) assumed to represent 3-sigma uncertainty. To ensure that sterol was not over-represented as two independent variables in the subsequent principal component analysis, we employed fractional sterol, which is sterol flux times sterol ratio. Similarly, for the fourth constituent series, lake diatoms, we used a weighed sum to combine the two principle components (resulting from analyses carried out on centred and standardised, square root transformed relative abundance data) into a single index. This weighed sum is proportional to the per cent variance explained by each principal component (34% for PC1 and 27% for PC2) and assumed an uncertainty equivalent to 10%.

We performed separate principal component analyses on the median time series of the three constituent marine time series and the four constituent lake time series and only considered the resulting first principal component. We propagated the Monte Carlo uncertainty envelopes as fractional mean uncertainties bounding the first principal components of both the marine and lake records (Supplementary Table 2). In the marine record, Diatoms and *Chaetoceros* spores load positively on the first principal component (PC1 loadings of 0.63 and 0.62, respectively), while HBI III loads negatively (PC1 loading of −0.47). We interpret this as suggesting that the first marine principal component is positively correlated with primary production. In the lake record, all-time series (Cd:Ti, $\delta^{15}$N, sterols and diatoms) load positively on the first principal component (Supplementary Table 2). We interpret this as suggesting that the first lake principal component is positively correlated with little auk abundance. We do not consider principal components beyond the first but acknowledge that they may also contain climate and/or population signals.

When plotting both the first marine and lake principal components (Fig. 4), we included the respective Monte Carlo uncertainty envelopes, which have been propagated as the fractional mean uncertainties of the constituent time series

underlying each principal component. The Y-envelope corresponds to uncertainty in principal component magnitude, while the X-envelope corresponds to uncertainty in time. The temporal uncertainties we discuss are associated with these latter horizontal uncertainty envelopes.

*General additive models for detection of significant changes*. To identify points of significant change in our proxy data over time, we used general additive models (GAM), and took a flexible approach to the degree of smoothing employed in modelling the response of both the marine and lake record proxies. Where possible, the degree of smoothing was estimated using restricted maximum likelihood (REML), and the model contained a continuous-time first-order autoregressive process (CAR(1)) to account for temporal autocorrelation. This solution was optimal for the marine record proxies Diatoms, *Chaetoceros* spores, IP$_{25}$ and HBI III, and for $\delta^{15}$N and the diatom PC1 and PC2 axes scores from the lake record. However, for Cd:Ti and cholesterol from the lake record, this approach produced an over-smoothed model, and so in these cases, we used a generalised cross validation (GCV) approach, which optimises predictive accuracy and allows for heteroscedasticity in the data. After the model was fitted (Supplementary Figs. 7–15), posterior simulation was run with 20 random draws from the posterior distributions of the fitted GAMs to reflect the degree of certainty of the model at a given time (Supplementary Figs. 7–15), and to allow for the calculation of confidence intervals. Finally, first derivatives (black lines) and associated 95% confidence intervals were estimated for the GAM trend of each proxy. Using this approach, a significant change is demonstrated when the confidence interval of the first derivative does not include 0 (Supplementary Figs. 7–15)[68]. GAM analyses were carried out in R version 4.0.0[69], using the package mgcv[70,71] and additional code outlined in ref. [68].

## Data availability
Proxy data are available at the open data repository of the Geological Survey of Denmark and Greenland (GEUS Dataverse https://doi.org/10.22008/FK2/MQDS1L)[72]. The historical air temperature data from Pituffik is available from the Danish Meteorological Institute (dmi.dk), and Arctic Oscillation index data from NOAA/CPC.

## Code availability
Code used for numerical analyses in this study is available at the open data repository (GEUS Dataverse https://doi.org/10.22008/FK2/MQDS1L)[72] of the Geological Survey of Denmark and Greenland.

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

## Acknowledgements
This study received financial support from EU's FP7 project ICE-ARC under Grant Agreement No. 603887, the North Water Project (www.NOW.KU.DK) funded by the Velux Foundations and the Carlsberg Foundation of Denmark, the Villum Foundation Young Investigator programme (Grant VKR023454 to S.R.), and project GreenEdge. J.S. was funded by the Academy of Finland (project 296918), SR received support from the Independent Research Fund of Denmark (grant 9064-0039B) and A.L. was funded by the Natural Sciences and Engineering Research Council of Canada (Grant 2018-03984). We thank the people of Avanersuaq, and all participants of the ArcticNet 2015 Leg 4a expedition onboard CCGS *Amundsen*, especially Philippe Archambault. Lewis Collins is acknowledged for discussions and Kirsten Hastrup for hosting the first meeting that ultimately led to this interdisciplinary study. In memoriam of our dear colleague Lene K. Holm. We acknowledge the use of imagery from the NASA Worldview application (https://worldview.earthdata.nasa.gov/), part of the NASA Earth Observing System Data and Information System (EOSDIS).

## Author contributions
S.R. wrote the manuscript with input from all co-authors. S.R. and T.A.D. conceived the study, and respectively coordinated and synthesised data from the marine and lake studies. A.L. generated the diatom and biomarker data and prepared the figures together with S.R., R.J., T.A.D. and W.C. G.M., J.G. and X.C. provided the marine sediment core material, and contributed with laboratory analyses. S.R., A.L., G.M., J.G., X.C., K.W., E.G., N.M., A.K. and S.M.O. all contributed to the interpretation of the marine data. T.J.A. generated the $^{210}$Pb/$^{137}$Cs data and geochronology. S.M.O. provided oceanographic context to the interpretations. W.C. and T.A.D. performed the statistical analyses. J.O. and A.S. conducted the $^{14}$C dating, J.S. contributed isotope data, A.C.G.H., H.M. and S.T. generated the sterol data, N.K.L. oversaw the XRF data, K.L.J. and A.M. initiated the seabird aspects of the wider project and K.L.J. commented extensively on the manuscript. S.W. contributed and synthesised data on the landscape development of the lake site. E.J., B.G. and M.N. contributed with expert knowledge on the local ecology (E.J.), archaeology (B.G.) and anthropology (M.N). All co-authors commented on the manuscript and approve its content.

## Competing interests
The authors declare no competing interests.
