## [Peer Review File · Nature Communications]

Editorial Note: This manuscript has been previously reviewed at another journal that is not operating a transparent peer review scheme. This document only contains reviewer comments and rebuttal letters for versions considered at Nature Communications. Mentions of the other journal have been redacted.

REVIEWER COMMENTS

Reviewer #1 (Remarks to the Author):

After having read carefully the new version, I find that the authors did a great work in responding the questions raised.

One important aspect of this paper should be highlighted: the recent changes in sea ice proxies. As of now, Figure 4d (IP-25 and HPI-III) shows the sea ice with the earliest date at 2000CE. Because the Arctic has been warming at unprecedented rate after ~2005, it would be interesting if you add the period ~2000CE-2015CE (or the most recent date), i.e. combining Figure 7f with Figure 4d, to get the full coverage and a better picture as to how the recent decades compare to the past several millennia. Perhaps something to add in the supplements.

I also looked for the Pb-210 and Cs-137 data and did not find them in the MS. It is possible that no apparent Cs-137 peak was detected. If detected, this would give more credibility to the interpretation in Figure 7f, as you compare your data to modern climatology. I suggest that a Supplementary Figure showing the Pb-210 and the Cs-137 (total activity, supported, unsupported, Cs fluxes) be added.

Following these rather small comments, I recommend publication of this manuscript in Nature Communications.

Here are some minor comments:

- Maybe add references to the 5th line of your introduction.
- In methods: Add info for the CT-scan: where and what settings
- Unequivocal is used 4 times, maybe use a synonym for one of them
- In Polynya Instability (6th line): the earlier decline could be linked to different location, but also one cannot rule out chronological uncertainties. It would be fair to point this out.
- Figure S2d-S9d: add a bold horizontal line at the zero line to better see what is significant in the "First derivative plot"
- Figure S7d : Significant changes are observed for the period before 4000, 2000 and 1000B2K, but the one after 1000K in Fig. S7d does not seem to be significant. In Fig. 4g, a star is seen after 1000b2K in the Cd/Ti. Is this really significant?

Reviewer #2 (Remarks to the Author):

In this manuscript, past changes in sea ice cover and primary production in the North Water polynya, as well as the population dynamics of little auk colonies in a lake in Greenland were reconstructed using a multi-proxies analysis of marine and lake sediment records. These ecosystem changes are subsequently combined with published evidence on the presence of humans in Greenland during the past c. 4500 years. I have been asked to specifically review the interpretation of lacustrine sediment record in terms of lake evolution and whether the marked shift in sedimentology might reflect a lake isolation event rather than the colonization of the catchment by little auk. This issue was raised by reviewer #2 in a previous report.

Based on the diatom species composition, it can be concluded that the sediments before the marked change in sedimentology were deposited in a lacustrine environment. I fully agree with the authors that the diatoms dominating the bottom parts of the core are freshwater taxa. Hence, the major transition in the lacustrine sediment core is likely not related to the lake being isolated from the sea. However, I do have a number of (major) comments which need to be dealt with before the manuscript can be accepted.

First, instead of summarizing the major diatom taxa in a table (Table S2), I highly recommend to provide a stratigraphic plot of e.g. the 20-30 most abundant diatom taxa in the lake sediment core. The authors can indicate the autoecology of the taxa by using accolades above the species names. No use of indicating whether the species are freshwater (all of them are), but it might be useful to indicate species with a lower/higher optimum for pH and/or nutrient concentrations (see third comment). This figure can replace the table in the supplementary material. A similar diagram should be produced for the ITRAX data. Now it seems that the authors have only plotted selected elements from the ITRAX data. I strongly recommend that the authors make a stratigraphic plot of all major elements instead of cherry-picking a few of them (Roberts et al 2017 Nature Communications DOI: 10.1038/ncomms14914 is a good example). This was also suggested by reviewer #4 by the way. I also believe that a diatom stratigraphic plot would be useful for the marine sediment core. I am fully aware that papers in high impact journals need to be concise, but at the same time, I strongly believe that all data used in such papers need to be visualized in an appropriate way. This ensures full transparency.

Second, the rationale given based on existing regional RSL data for the region for motivating that the transition in the core is erroneous as they don't exclude an isolation event at 4200 yr BP. The data presented in Lecavalier et al. (2014) are raised beach data of the minimum and maximum height of the relative sea level (white and grey triangles in the figure provided in rebuttal letter 2). Because the geological constraints of RSL in the two sites near the study area are based on raised beach data, there is no information on the exact height of RSL after c. 8000 yr BP. Raised beached data provide a minimum height of RSL in a region. The RSL curves in Lecavalier et al. (2014) are based on a modelling experiment and not on geological constraints. In fact, the RSL curves were tested against the geological constraints

and in some regions (and particularly North-west Greenland) the models are not performing that well as discussed by these authors. It follows that these RSL curves presented in Lecavalier et al. (2014) cannot be used to rule out that the lake was situated above sea level at c. 4200 cal yr BP and became isolated earlier. Hence, I strongly recommend that the authors focus their interpretation on their own data (i.e. the presence of freshwater diatoms and low Cl concentrations) rather than 'misusing' the modelling results from Lecavalier et al. (2014). What is certain is that the study lake is an isolation basin, but when it became isolated is (unfortunately) not possible to derive from the present lake sediments.

Third, I am convinced that the full potential of the diatom data in the lake sediment core has not been fully grasped. As correctly pointed out by the authors in one of their rebuttal letters, diatoms are indeed very powerful bioindicators. This is not only the case for changes in salinity, but also to reconstruct changes in nutrient concentrations and pH (see the seminal work by John Smol, Richard Battarbee and others). I am not an Arctic diatom specialist, but I am aware of several calibration datasets (Pienitz & Cournoyer 2017 JoPL and the work of Dermot Antoniades for the Canadian Arctic), which can be used to calculate the optimum for pH and nutrient (P?) concentrations. I was quite surprised that TP wasn't analysed by the way. This information is available from the ITRAX dataset I assume? As a very minimum, the authors can give an indication of diatom indicator taxa. Without knowing the autoecology of the species present in the lacustrine sediment core, it is quite obvious that the lake evolved from a relatively neutral system to a more acidic environment (as evidenced by the presence of the *Eunotia* taxa in the zone presumably influenced by marine birds). This is very likely related to the development of extensive peat banks in the catchment in response to nutrient additions (and marine birds being present...). The taxa in the bottom part of the core are *Fragilarioid* taxa and hence characteristic for Arctic lakes influenced by glacial meltwater (pointed out by reviewer 2 – work done by Bianca Perren and Alexander Wolf might be worth to check). I strongly recommend that the authors thoroughly interpret the changes in diatom community composition in the light of changes in e.g. pH and nutrient concentrations. This will provide an independent proxy for the inferred changes in population dynamics in little auk communities. And it will make the paper much stronger.

Some minor comments:

Psammothidium in Table S2 should be *Psammothidium*. I also recommend to use regional studies for assessing the salinity optimum and habitat type (benthic/planktonic). There are a lot of studies available from the Arctic!

Just to be certain: 'Diatoms' in Fig.4b are all diatoms, excluding *Chaetoceros* resting spores? If so (and it should be), I suggest to change the axis title to 'Diatoms (excluding *Chaetoceros* RS)'.

Were the principal component analyses done on standardized and centred data? This should be the case because variables expressed in different units were analysed together. Also, why don't include the diatom data in the lacustrine sediment core (see comment 3)?

In summary, an interesting study which, however, requires some more work to integrate the impressive set of proxies analysed in a marine and lacustrine sediment record from Greenland.

Response to reviewers

Reviewer #1 (Remarks to the Author):

After having read carefully the new version, I find that the authors did a great work in responding the questions raised.

One important aspect of this paper should be highlighted: the recent changes in sea ice proxies. As of now, Figure 4d (IP-25 and HPI-III) shows the sea ice with the earliest date at 2000CE. Because the Arctic has been warming at unprecedented rate after ~2005, it would be interesting if you add the period ~2000CE-2015CE (or the most recent date), i.e. combining Figure 7f with Figure 4d, to get the full coverage and a better picture as to how the recent decades compare to the past several millennia. Perhaps something to add in the supplements.

We agree with this suggestion that it is useful to show the most recent changes in sea ice proxies in combination with the record of the past millennia. We have decided to add IP25 and HBI III concentrations normalized by TOC to Figure 2, which presents information on the composite record. This way, the reader can directly compare changes in sedimentation rate and TOC content with changes in sea ice biomarker concentrations for the entire period.

I also looked for the Pb-210 and Cs-137 data and did not find them in the MS. It is possible that no apparent Cs-137 peak was detected. If detected, this would give more credibility to the interpretation in Figure 7f, as you compare your data to modern climatology. I suggest that a Supplementary Figure showing the Pb-210 and the Cs-137 (total activity, supported, unsupported, Cs fluxes) be added.

This is a good point. In addition to presenting the ^{210}Pb chronology for the box core in Figure 2, we have now added a Supplementary Figure showing the ^{210}Pb and ^{137}Cs data as suggested (new Supplementary Fig. 1).

We also added the following text to the methods: “In the box core, the content of unsupported ^{210}Pb showed a clear exponential decline with depth. A clear ^{137}Cs peak was not detected, but the ^{210}Pb -based chronology dates the earliest sample with ^{137}Cs to 1969 +/- 2 years, which is close to the expected date, 1963, for the global ^{137}Cs peak induced by nuclear weapons testing in the atmosphere. This, and the very uniform exponential decline in unsupported ^{210}Pb with depth, gives confidence in the calculated chronology.”

Following these rather small comments, I recommend publication of this manuscript in Nature Communications.

We thank the reviewer for the positive and constructive evaluation of our work and for these additional remarks and helpful suggestions.

Here are some minor comments:

- Maybe add references to the 5th line of your introduction.

We have now added an additional reference to support this statement.

- In methods: Add info for the CT-scan: where and what settings

We have added the following text to the methods: “Computed Tomography (CT) scanning was performed using a Siemens SOMATOM Definition AS+ 128 at the Institut National de la Recherche Scientifique (INRS), Quebec, Canada. The tomograms were converted into digital DICOM format using a standard Hounsfield scale (HU scale) from -1024 to 3071, where -1024 corresponds to the density of air, 0 to the density of water and 2500 to the density of calcite”.

- Unequivocal is used 4 times, maybe use a synonym for one of them

We have followed the suggestion and replaced this word to avoid repetition, thank you.

- In Polynya Instability (6th line): the earlier decline could be linked to different location, but also one cannot rule out chronological uncertainties. It would be fair to point this out.

Agreed. We have revised the sentence and it now reads: “Although dating uncertainties cannot be ruled out as an alternative explanation, this idea is supported by peat core studies from further north, demonstrating that bird colonies spread northwards around 2800 and 2200 cal yrs b2K”.

- Figure S2d-S9d: add a bold horizontal line at the zero line to better see what is significant in the “First derivative plot”

We have added a horizontal line marking “zero” on all the GAMs figures as suggested.

- Figure S7d : Significant changes are observed for the period before 4000, 2000 and 1000B2K, but the one after 1000K in Fig. S7d does not seem to be significant. In Fig. 4g, a star is seen after 1000b2K in the Cd/Ti. Is this really significant?

Yes, the GAMs analyses detected a point of significant change in Cd/Ti at about 900b2k.

Reviewer #2 (Remarks to the Author):

In this manuscript, past changes in sea ice cover and primary production in the North Water polynya, as well as the population dynamics of little auk colonies in a lake in Greenland were reconstructed using a multi-proxies analysis of marine and lake sediment records. These ecosystem changes are subsequently combined with published evidence on the presence of humans in Greenland during the past c. 4500 years. I have been asked to specifically review the interpretation of lacustrine sediment record in terms of lake evolution and whether the marked shift in sedimentology might reflect a lake isolation event

rather than the colonization of the catchment by little auk. This issue was raised by reviewer #2 in a previous report.

Based on the diatom species composition, it can be concluded that the sediments before the marked change in sedimentology were deposited in a lacustrine environment. I fully agree with the authors that the diatoms dominating the bottom parts of the core are freshwater taxa. Hence, the major transition in the lacustrine sediment core is likely not related to the lake being isolated from the sea. However, I do have a number of (major) comments which need to be dealt with before the manuscript can be accepted.

First, instead of summarizing the major diatom taxa in a table (Table S2), I highly recommend to provide a stratigraphic plot of e.g. the 20-30 most abundant diatom taxa in the lake sediment core. The authors can indicate the autoecology of the taxa by using accolades above the species names. No use of indicating whether the species are freshwater (all of them are), but it might be useful to indicate species with a lower/higher optimum for pH and/or nutrient concentrations (see third comment). This figure can replace the table in the supplementary material. A similar diagram should be produced for the ITRAX data. Now it seems that the authors have only plotted selected elements from the ITRAX data. I strongly recommend that the authors make a stratigraphic plot of all major elements instead of cherry-picking a few of them (Roberts et al 2017 Nature Communications DOI: 10.1038/ncomms14914 is a good example). This was also suggested by reviewer #4 by the way. I also believe that a diatom stratigraphic plot would be useful for the marine sediment core. I am fully aware that papers in high impact journals need to be concise, but at the same time, I strongly believe that all data used in such papers need to be visualized in an appropriate way. This ensures full transparency.

We thank the reviewer for the positive and constructive evaluation of our work, and for confirming that the lake diatom taxa are freshwater.

We have followed the reviewer's suggestion and now provide an additional plot including all the major elements from the ITRAX analyses as supplementary information (new Supplementary Figure 2). We would like to mention, however, that the decision to include only selected elements such as Cl in our previous submission was justified by the fact that we were specifically asked to provide XRF-based evidence for freshwater deposition. But we fully agree with the reviewer that it is useful to provide supplementary information on the other elements as well and have now done so.

Following the reviewer's suggestion, we now also provide a detailed stratigraphic diagram for the lake diatom assemblages (new Supplementary Fig 3) and have performed new numerical analyses and interpretation of the lacustrine diatom assemblage composition record, primarily in the light of changes in pH (see also response to third comment). While an interpretation of ecological changes in the lake is beyond the scope of this study, we fully agree with the reviewer that the addition of these data strengthens the manuscript in the sense that they provide additional evidence and support to our interpretations on little auk colony dynamics (and thus polynya stability and productivity changes - the focus of the study).

However, we respectfully disagree with the reviewer's comment "I also believe that a diatom stratigraphic plot would be useful for the marine sediment core". Marine diatoms, including

Chaetoceros spores, are used in this study as proxies for marine primary production, as it is well known that diatoms dominate the spring/summer bloom in the North Water polynya and their fluxes indicate the magnitude and duration of the phytoplankton bloom. The source data are thus not species composition data, but total diatom (including *Chaetoceros*) counts and fluxes. We did analyse the taxonomic composition of the marine diatoms present, but these analyses were performed as a parallel study for a separate project aimed at quantitative reconstructions of regional sea-surface temperatures in northern Baffin Bay using transfer functions. The assemblage data and transfer function work represent more than 1 year of full-time work by an ECR and it is in preparation to be published elsewhere. We understand (particularly the three polar diatomists among the co-authors) the curiosity of wanting to see additional species data, but in this case we believe adding such a dataset as supplementary information would not serve the story, as its interpretation would be outside the scope of this study, and would instead jeopardize another project (due to dual publication issues). We hope the reviewer understands our position.

In summary, we have revised the manuscript to include a discussion on the lake diatom assemblage composition and how it reflects changes in the lake depositional environment over time. We have also replaced the Supplementary species name list with Supplementary Fig. 3 showing a stratigraphic plot of the main species, and added a Supplementary Figure (Supp. Fig. 2) showing all the major XRF elements. The results of the PCA analyses on the lake diatom assemblage are plotted in Fig. 4 (PCA axes 1 and 2) and we have further performed GAM analyses to derive points of significant change (shown in Supplementary Figs. 14 and 15). We performed new Monte Carlo simulations and principal component analyses for the lake record (Supplementary Fig 6) to fully integrate the diatom assemblage data. We note that integrating these data did not change the main findings of the study (resulting lake PC shown in Fig4), but it does offer additional support to our interpretations.

Second, the rationale given based on existing regional RSL data for the region for motivating that the transition in the core is erroneous as they don't exclude an isolation event at 4200 yr BP. The data presented in Lecavalier et al. (2014) are raised beach data of the minimum and maximum height of the relative sea level (white and grey triangles in the figure provided in rebuttal letter 2). Because the geological constraints of RSL in the two sites near the study area are based on raised beach data, there is no information on the exact height of RSL after c. 8000 yr BP. Raised beached data provide a minimum height of RSL in a region. The RSL curves in Lecavalier et al. (2014) are based on a modelling experiment and not on geological constraints. In fact, the RSL curves were tested against the geological constraints and in some regions (and particularly North-west Greenland) the models are not performing that well as discussed by these authors. It follows that these RSL curves presented in Lecavalier et al. (2014) cannot be used to rule out that the lake was situated above sea level at c. 4200 cal yr BP and became isolated earlier. Hence, I strongly recommend that the authors focus their interpretation on their own data (i.e. the presence of freshwater diatoms and low Cl concentrations) rather than 'misusing' the modelling results from Lecavalier et al. (2014). What is certain is that the study lake is an isolation basin, but when it became isolated is (unfortunately) not possible to derive from the present lake sediments.

We agree that the evidence from Thule and Saunders Island are based on radiocarbon dating of molluscs found in raised marine deposits and archeological remains of pre-Inuit groups. We also agree that the RSL history would be improved if index points from e.g. isolation lakes existed from the study

area, but unfortunately, no such data are available. However, we believe that the existing geological information and modelling results indicate that the *most likely* RSL history from the area precludes that the lake was isolated as late as 4200 yrs BP. The model results presented in Lecavalier et al (2014) are to a large degree supported by the geological data suggesting that the RSL dropped from the marine limit at c. 40 to 0 m a.s.l. between c. 11,000 and 8000 yrs BP i.e. much earlier than 4200 yrs BP. Also, if the RSL in the study area remained above 8 m a.s.l. (where the lake is located today) then it would be expected that some of the dated molluscs would fall in the range from 8000-4200 yrs BP. None of the dated molluscs fall within that interval and that suggests that this is not a likely interpretation. This approach to interpreting RSL data, where the RSL curve is placed above the youngest radiocarbon ages of the raised marine deposits, has been used in several studies around Greenland (e.g. Hall et al., 2010; Funder et al., 2011). In addition, if the RSL should remain above 8 m a.s.l. until after 4200 yrs BP it would require that the Greenland Ice Sheet had a larger extent than present until then, to account for the isostatic depression below sea level. This scenario is incompatible with the existing reconstructions of the Greenland Ice Sheet both Greenland-wide (e.g. Larsen et al., 2015; Briner et al., 2015) and more specifically in NW Greenland, where geological observations demonstrate that local glaciers and the GrIS were smaller than present from 6400 to 400 yrs BP, as a response to the HTM (e.g. Farnsworth et al., 2018; Søndergaard et al., 2019, 2020). In summary, we think it is reasonable to argue that the *most likely* RSL history precludes that the transition at 4200 yrs BP in the lake core represents a marine isolation.

However, we acknowledge Reviewer #2's suggestions and agree to focus our interpretation on the data from our lake record, since we have overwhelming evidence (diatom assemblages and XRF data) of the record representing a continuous freshwater depositional environment. We have thus revised the text accordingly, and now place emphasis on our data.

Third, I am convinced that the full potential of the diatom data in the lake sediment core has not been fully grasped. As correctly pointed out by the authors in one of their rebuttal letters, diatoms are indeed very powerful bioindicators. This is not only the case for changes in salinity, but also to reconstruct changes in nutrient concentrations and pH (see the seminal work by John Smol, Richard Battarbee and others). I am not an Arctic diatom specialist, but I am aware of several calibration datasets (Pienitz & Cournoyer 2017 JoPL and the work of Dermot Antoniades for the Canadian Arctic), which can be used to calculate the optimum for pH and nutrient (P?) concentrations. I was quite surprised that TP wasn't analysed by the way. This information is available from the ITRAX dataset I assume? As a very minimum, the authors can give an indication of diatom indicator taxa. Without knowing the autoecology of the species present in the lacustrine sediment core, it is quite obvious that the lake evolved from a relatively neutral system to a more acidic environment (as evidenced by the presence of the *Eunotia* taxa in the zone presumably influenced by marine birds). This is very likely related to the development of extensive peat banks in the catchment in response to nutrient additions (and marine birds being present...). The taxa in the bottom part of the core are *Fragilarioid* taxa and hence characteristic for Arctic lakes influenced by glacial meltwater (pointed out by reviewer 2 – work done by Bianca Perren and Alexander Wolf might be worth to check). I strongly recommend that the authors thoroughly interpret the changes in diatom community composition in the light of changes in e.g. pH and nutrient concentrations. This will provide an independent proxy for the inferred changes in population dynamics in little auk communities. And it will make the paper much stronger.

We thank the reviewer for these suggestions. We have now performed numerical analyses of the diatom assemblage composition data and performed GAM analyses independently for each of the two first PCA axes to determine points of significant change in the diatom assemblage record (Supp figures 3, 6, 14 and 15). The two main PCA axes are now plotted in Figure 4 as additional lake proxies and stars mark points in time where significant changes occur, based on the GAM results. Indeed, the diatom assemblages clearly track changes in the population dynamics of little auks over time and give additional and independent support to our interpretation of the lake record. We carefully considered the autecology of the species present in the lake, besides their freshwater affinity, and conclude that they primarily indicate changes in pH. As the reviewer rightly points out, this is likely related to peat development in the catchment, fueled by marine-derived nutrients from little auk guano (extensive peat development is also seen in Supplementary Fig 16). Despite the high N loading evidenced by our other proxies, there are no indicator species of eutrophication in the diatom assemblages. This is not too surprising considering that the lake is ice-covered for about 10 months/year and planktic species are very rare in this record. Known indicators of a higher trophic state are typically planktic species but these, however, do not thrive in acidic waters. We have revised the Results and Discussion section to include the main results and interpretations from the lake diatom data as suggested.

Some minor comments:

Psammothidium in Table S2 should be Psammothidium. I also recommend to use regional studies for assessing the salinity optimum and habitat type (benthic/planktonic). There are a lot of studies available from the Arctic!

We have corrected this, thank you. Additional references on regional studies have also been added.

Just to be certain: ‘Diatoms’ in Fig.4b are all diatoms, excluding Chaetoceros resting spores? If so (and it should be), I suggest to change the axis title to ‘Diatoms (excluding Chaetoceros RS)’.

Yes. We have revised the axis label as suggested.

Were the principal component analyses done on standardized and centred data? This should be the case because variables expressed in different units were analysed together. Also, why don’t include the diatom data in the lacustrine sediment core (see comment 3)?

Yes, we used the MATLAB “pca.m” function which, by default, centers all data. And yes, we normalized all series, to account for differences in mean and standard deviation, prior to inputting them to pca.m. And, as stated earlier, we have now included the diatom data in the Monte Carlo simulations and PC compilation for the lake record proxies.

In summary, an interesting study which, however, requires some more work to integrate the impressive set of proxies analysed in a marine and lacustrine sediment record from Greenland.

We thank the reviewer for the extensive comments, which have helped us to substantially improve the study, especially the analysis of the lake record.

REVIEWERS' COMMENTS

Reviewer #2 (Remarks to the Author):

After reading the rebuttal letter and the revised manuscript, I believe that the authors have dealt with all my comments made on the previous version of the manuscript. Frankly speaking, I am quite happy that the authors agreed to include a stratigraphic plot of the lake diatoms. This made the paper definitely stronger.

I have no further comments. Interesting study.

NCOMMS-21-06730A - Ribeiro et al

REVIEWERS' COMMENTS

Reviewer #2 (Remarks to the Author):

After reading the rebuttal letter and the revised manuscript, I believe that the authors have dealt with all my comments made on the previous version of the manuscript. Frankly speaking, I am quite happy that the authors agreed to include a stratigraphic plot of the lake diatoms. This made the paper definitely stronger.

I have no further comments. Interesting study.

RESPONSE

We thank the reviewer for their positive evaluation of our work and for the constructive comments and suggestions that helped improve the manuscript.